# The Impact of Postoperative Complications on Survival after Simultaneous Resection of Colorectal Cancer and Liver Metastases

**DOI:** 10.3390/healthcare10081573

**Published:** 2022-08-19

**Authors:** Sorin Tiberiu Alexandrescu, Narcis Octavian Zarnescu, Andrei Sebastian Diaconescu, Dana Tomescu, Gabriela Droc, Doina Hrehoret, Vladislav Brasoveanu, Irinel Popescu

**Affiliations:** 1Department of General Surgery, Carol Davila University of Medicine and Pharmacy, 050474 Bucharest, Romania; 2Department of Surgery, Center for Digestive Disease and Liver Transplantation, Fundeni Clinical Institute, 022328 Bucharest, Romania; 3Second Department of Surgery, University Emergency Hospital Bucharest, 050098 Bucharest, Romania; 43rd Department of Anesthesia and Intensive Care, Fundeni Clinical Institute, 022328 Bucharest, Romania; 51st Department of Anesthesia and Intensive Care, Fundeni Clinical Institute, 022328 Bucharest, Romania; 6Faculty of Medicine, « Titu Maiorescu » University, 040441 Bucharest, Romania

**Keywords:** liver metastases, colorectal cancer, simultaneous resection, survival, postoperative complications

## Abstract

Background: The aim of this study was to investigate the impact of postoperative complications on the long-term outcomes of patients who had undergone simultaneous resection (SR) of colorectal cancer and synchronous liver metastases (SCLMs). Methods: We conducted a single-institution survival cohort study in patients with SR, collecting clinical, pathological, and postoperative complication data. The impact of these variables on overall survival (OS) and disease-free survival (DFS) was compared by log rank test. Multivariate Cox regression analysis identified independent prognostic factors. Results: Out of 243 patients, 122 (50.2%) developed postoperative complications: 54 (22.2%) major complications (Clavien–Dindo grade III–V), 86 (35.3%) septic complications, 59 (24.2%) hepatic complications. Median comprehensive complication index (CCI) was 8.70. Twelve (4.9%) patients died postoperatively. The 3- and 5-year OS and DFS rates were 60.7%, 39.5% and 28%, 21.5%, respectively. Neither overall postoperative complications nor major and septic complications or CCI had a significant impact on OS or DFS. Multivariate analysis identified the N2 stage as an independent prognostic of poor OS, while N2 stage and four or more SCLMs were independent predictors for poor DFS. Conclusion: N2 stage and four or more SCLMs impacted OS and/or DFS, while CCI, presence, type, or grade of postoperative complications had no significant impact on long-term outcomes.

## 1. Introduction

Colorectal cancer is the most frequent digestive cancer, accounting for 10% of all cancer diagnoses and 10% of cancer-related deaths worldwide [1]. Prognosis depends on clinical stage, although recent studies revealed that molecular and genetic features, as well as postoperative complications, may impact long-term outcomes. Approximately one quarter of colorectal patients are diagnosed with synchronous liver metastasis (SCLMs), while 30% develop liver metastases subsequently (metachronous metastases) [2]. Among the genetic mechanisms involved in the progression of colorectal cancer, Bcl-2–Bax binding sems to play an important role, through the inhibition of apoptosis. Thus, Bax mutations would be a target to prevent the progression of primary colorectal cancers to the metastatic stage [3].

Distant metastases represent the major cause of cancer-related death [4]. Although patients with SCLMs have worse prognosis compared to patients with metachronous liver metastases, [5,6,7] 5-year overall survival (OS) rates higher than 30% could be achieved in such patients in most high-volume centers. A major challenge in the surgical treatment of colorectal cancer with synchronous liver metastases is deciding on the timing of primary tumor resection and hepatectomy, respectively.

In patients with SCLMs, three surgical approaches are available. The classic approach is resection of the primary colorectal tumor, followed at a later point in time by resection of liver metastases (delayed liver resection). A newer approach involves the simultaneous resection (SR) of colorectal cancer and SCLMs [8,9,10,11]. A third approach involves initial hepatectomy and subsequent colorectal resection (“liver-first” approach) [12].

A major concern regarding SR is the higher incidence of postoperative complications and their potential impact on long-term outcomes, and this has led to reluctance to perform SR [13,14]. Despite the attractiveness of the concept of the simultaneous excision of colon cancer and liver metastases, there is still a paucity of high-quality evidence. This study will evaluate the outcomes of simultaneous colorectal and liver resection utilizing a pragmatic approach and rigorous patient selection in a high-volume center for both procedures.

## 2. Materials and Methods

Liver metastases were classified as synchronous if they were diagnosed either during the preoperative evaluation or during surgery for the primary colorectal tumor. A detailed description of standard preoperative work-up was previously provided [10].

### 2.1. Study Design and Population

We conducted a single-institution survival cohort study in patients with SR, collecting clinical, pathological, and postoperative complication data. All consecutive adult patients who underwent SR between January 2002 and December 2018 were retrieved from a prospectively maintained database including all the patients who underwent resection of CLMs in our center. Patients with extrahepatic metastases, as well as those with incomplete follow-up data, were excluded from this analysis. The flow-chart diagram of the included patients is presented in Figure 1.

### 2.2. Treatment Allocation

Patients with SCLMs were usually evaluated by a multidisciplinary team (including oncologists, radiologists, as well as colorectal and hepatic surgeons). The multidisciplinary team has recommended the timing of multimodal treatment and surgical team selected patients for surgical procedure.

The decision to perform SR or staged resection was made by the attending surgeon based on preoperative and intraoperative findings (complexity of the hepatectomy and colectomy), the performance status of the patient, and the anticipated technical ability to perform a complete resection of the primary tumor and SCLMs with preservation of more than 30% of total liver volume. In line with the current practice, the policy of the center is to perform SR in patients with uncomplicated primary tumors (no bowel occlusion or perforation) and requiring minor hepatectomies [15,16]. Major liver resections are considered only in patients with very good performance status and who have not undergone a difficult colorectal procedure.

As a principle, patients with severe medical comorbidities were excluded from SR. In recent practice survey, patient comorbidities were ranked as the most significant barrier by 35% of surgeons [16]. Furthermore, performance status is used to guide patient selection for SR [15].

### 2.3. Data

The following data points were collected and evaluated: age, gender, location of the primary tumor, pathology data of colorectal tumor and liver metastases (pT, pN, maximum metastases diameter, number of SCLMs), tumor burden score (TBS), uni/bilobar distribution of SCLMs, the use of neo-adjuvant and adjuvant oncologic therapy, as well as the presence, type, and grade of postoperative complications and comprehensive complication index (CCI). TBS was calculated according to the formula: TBS^2^ = (maximum tumor diameter)^2^ + (number of tumors)^2^, as previously published [17].

### 2.4. Postoperative Complications

Postoperative complications were graded according to the Clavien–Dindo classification [18]. Major complications were defined as Clavien–Dindo Grade III or higher. In patients who developed more than one complication, the highest grade of complications was utilized in analysis. For a more comprehensive evaluation of the overall impact of postoperative complications on long-term outcomes, CCI was evaluated for each patient. Postoperative mortality was defined as the death of the patient during hospital stay or within the first 30 days after surgery. Hepatic complications included liver cut-surface collection, biliary fistula, and liver failure. Septic complications included liver cut-surface collections, anastomotic fistula, pelvic abscess, partial bowel necrosis, wound septic complications, pneumonia, urinary tract infections, and Clostridium difficile infection.

### 2.5. Outcome Measurements

The OS was defined as the interval between the SR and the date of the patient’s death or the last follow-up (performed by personal contact with the patient, the patient’s family, or the attending oncologist). The disease-free survival (DFS) was calculated as the time between SR and the date of malignancy recurrence or the last follow-up, if the patient was free of disease at that moment.

The study protocol was approved by the local Institutional Review Board with the number 30768/11 June 2020.

### 2.6. Statistical Analysis

Categorical data are presented as absolute numbers and percentages. Association between categorical variables was analyzed with Fischer’s exact test. Fisher’s exact test is preferable because it is an exact test (while the chi-squared test relies on an approximation) and it is more suitable in analyses of small samples [19]. Continuous data are presented as mean (+/− standard deviation) or median and interquartile range (IQR), depending on the normality of the data distribution. Normality distribution for continuous data was assessed by Shapiro–Wilk test and further comparison was performed with t-test or Mann–Whitney U test, accordingly. Survival rates were estimated with the Kaplan–Meier method and were compared between different groups by log rank test and Cox proportional hazards test. In univariate analysis, we evaluated the impact of the following parameters on both OS and DFS: age, gender, primary tumor location, T stage, N stage, distribution, size, and number of SCLMs, TBS, type and extension of hepatectomy, neoadjuvant and adjuvant treatment, as well as presence, type, and severity of postoperative complications and CCI. Multivariate Cox proportional hazards regression analysis with backward stepwise selection process was used to identify the independent risk factors associated with DFS and OS. The multivariate analysis included those parameters that were associated with a *p* value less than 0.1 at univariate analysis. Hazard ratio (HR) was reported with 95% confidence interval (CI). A two-sided *p* value lower than 0.05 was considered significant. This statistical analysis was performed using IBM SPSS software, version 22 (SPSS Inc., Chicago, IL, USA).

The cut-off value of TBS able to divide the patients into two groups (low-TBS group vs. high-TBS group) with significantly different OS rates was determined by using X-tile software (Yale University, New Haven, CT, USA). The cut-off value of TBS was identified as the value that was associated with the maximum chi square (and the minimum *p* value).

## 3. Results

### 3.1. Study Cohort

Between January 2002 and December 2018, 292 patients with SCLMs underwent simultaneous colorectal and liver resection. Out of these patients, complete data regarding postoperative follow-up were available only in 243 cases (forming this study group). Of these 243 patients, there were 140 men (57.6%) and 92 patients older than 65 years of age. The primary tumor was located in the right colon in 48 cases (19.7%), left colon in 105 cases (43.2%), and rectum in 90 patients (37.1%). The majority of patients (n = 216, 88.8%) had T3 lesions. The rest of the cases were T1 in 1 patient (0.4%), T2 in 6 (2.4%), and T4 in 17 patients (7%). T status was missing in three patients (1.2%). Seventy-seven patients (31.7%) were N0 stage, 78 patients (32.1%) had N1 stage, 83 patients (34.2%) had N2 stage, and missing data were found in five patients (2%). Bilobar liver metastases were observed in 67 cases (27.5%) and hepatic lesions with a diameter above 3 cm in 117 cases (48.1%). The median size of liver metastases was 2.5 cm (IQR 1.5–4.5). Most of the patients had one liver lesion (n = 137, 56.3%), while 73 patients (30.1%) had two or lesions lesions and 33 patients (13.6%) had four or more lesions. The median TBS was 3.20 (IQR 2.19–5.40). Thirty-three patients (13.6%) had received neoadjuvant treatment, while 214 patients out of 231 patients who survived after surgery (92.6%) received postoperative oncologic therapy (Appendix A). Mean time from surgery to administration of adjuvant therapy was 6.74 (+/− 1.61) weeks.

The median level of CEA was 11.40 ng/mL (IQR 4.70–34.90). The preoperative level of CA 19-9 had a median value of 26.71 U/mL (IQR 12.39–103). Due to a high level of missing data (55%), these factors were not included in survival analysis.

### 3.2. Description of the Surgery

The type of colorectal procedure is presented in Table 1. Non-anatomic liver resections were performed in 209 patients (86%). Most patients (n = 211, 86.8%) had minor liver resection (less than three segments) and major liver resection was performed in 32 cases (13.2%). Right hepatectomy was performed in 11 cases, and it was associated with: left hemicolectomy (n = 5), subtotal colectomy (n = 2), right hemicolectomy (n = 1), abdominoperineal resection (n = 1), Hartmann operation (n = 1), and low anterior resection with protective ileostomy (n = 1). Left hepatectomy was performed in two cases, which were associated with Hartmann operation (n = 2). Right trisectionectomy was performed in two patients (associated with right colectomy), while left trisectionectomy was performed in two patients (associated with either low anterior resection with protective ileostomy or Hartmann operation). In 15 cases, other procedures resulting in major hepatectomy were associated with low anterior resection with protective ileostomy (n = 5), left hemicolectomy (n = 4), right hemicolectomy (n = 2), abdominoperineal resection (n = 2), Dixon operation (n = 1), and Hartmann procedure (n = 1).

### 3.3. Short-Term Outcomes

One hundred and twenty-two patients (50.2%) developed postoperative complications (Table 2). There were 68 patients (27.9%) with mild complications (Clavien–Dindo class I and II) and 54 patients (22.2%) with severe complications (Clavien–Dindo class III–V). Eleven patients developed two complications each. Thus, one patient with a hepatic septic complication (liver cut-surface collection) and concomitant anastomotic fistula evolved as a mild complication, and another patient with Clostridium difficile infection and biliary fistula evolved as a mild complication. Four patients with hepatic complications (two with biliary fistula and two with liver cut-surface collection) had a concomitant anastomotic fistula, evolving as severe complications, and one patient had both a biliary fistula and liver cut-surface collection, which also evolved as a severe complication. Another four patients that evolved as severe complications had two concomitant complications: acute pancreatitis and intra-abdominal abscess, anastomotic fistula and pneumonia, Clostridium difficile infection and liver cut-surface collection, cardiovascular and renal complications. Median CCI for the entire cohort was 8.70. Postoperative mortality rate was 4.9% (12 patients).

Fifty-nine patients (24.2%) had hepatic complications: 40 liver cut-surface collections, 18 biliary fistulas, and 1 transient liver failure. There was a statistically significant association between postoperative biliary fistula and type of hepatic resection (8.1% vs. 3% after non-anatomic vs. anatomic hepatectomy, respectively; *p* = 0.005). Major hepatic resection was also significantly associated with a higher rate of biliary fistula (16.7% vs. 6% after major vs. minor hepatectomy; *p* = 0.001).

Digestive anastomosis was performed in 184 patients and anastomotic fistula was recorded in 29 cases (15.7%). According to the location of the primary tumor that led to the anastomotic fistula, the leak was associated with rectal tumors in 10 patients (20.4%), left colon tumors in 15 patients (16.6%), and right colon cancers in 4 patients (8.3%). Major hepatectomy was marginally significantly associated with anastomotic fistula, compared to minor hepatectomy (29.1% vs. 13.7%; *p* = 0.0697). In all patients who underwent major liver resection, the hepatectomy was performed before the colorectal procedure and the Pringle maneuver was avoided in 58.9% of cases.

Eighty-six patients had septic complications: liver cut-surface collections (n = 40), anastomotic fistula (n = 29), abdominal/pelvic abscess (n = 6), pneumonia (n = 7), urinary tract infections (n = 3), wound septic complications (n = 2), Clostridium difficile infection (n = 3), and partial bowel necrosis (n = 1). Three patients with a hepatic septic complication had a concomitant anastomotic fistula. Two other patients had two septic complications (anastomotic fistula and pneumonia, Clostridium difficile infection, and liver cut-surface collection).

### 3.4. Long-Term Outcomes

The 12 patients who died within 30 days after surgery were excluded from the survival analysis. The 1-, 3-, 5-, and 10-year OS rates were 90.9%, 60.7%, 39.5%, and 21.1%, respectively, with a median OS of 47.9 months. After a median follow-up of 43.8 months, 159 patients developed recurrence of malignancy, while 72 were free of disease at the last follow-up. Subsequent repeat hepatectomy was performed in 39 patients (16%) with recurrent liver metastases. Median DFS was 16 months, with 1-, 3-, and 5-year DFS rates of 58.5%, 28%, and 21.5%, respectively.

### 3.5. Univariate Analysis

The factors that were associated with worse DFS in univariate analysis (*p* value < 0.05) were: N2 stage, bilobar distribution of SCLMs, the presence of four or more SCLMs, TBS higher than 3.2. Neoadjuvant chemotherapy and major liver resection were also associated with lower OS rates, although the difference was marginally significant (*p* value = 0.053 and 0.056, respectively) (Table 3).

The significant prognostic factors for poor OS in univariate analysis (*p* value < 0.05) were: N2 stage, the presence of four or more SCLMs, and preoperative chemotherapy. The patients with bilobar distribution of SCLMs had lower OS rates than those with unilobar metastases, although the difference was marginally significant in univariate analysis (HR = 1.391, 95% CI: 0.995-1.943; *p* value = 0.054) (Table 4).

The presence of postoperative complications, their grade or type, and CCI had no significant impact on either DFS or OS (*p* value > 0.05) (Figure 2, Table 3 and Table 4).

### 3.6. Multivariate Analysis

All the prognostic factors that were associated with a *p* value < 0.1 in univariate analysis were included in multivariate analysis. Multivariate Cox proportional hazards analysis identified N2 stage and the presence of four or more SCLMs as independent prognostic factors for poor DFS (Table 3). The only independent predictor of poor OS, based on multivariate Cox proportional hazards regression analysis, was N2 stage (Table 4).

## 4. Discussion

Although, during the last few decades, the resection of SCLMs has been widely accepted as the gold-standard treatment for these patients, a recent study revealed a lack of agreement on the timing of the surgical approach in patients with SCLMs, even among expert liver surgeons [20]. A major concern with SR for SCLMs is the possibility of a higher rate of postoperative complications vs. staged approaches (delayed liver resection or liver-first approach) [21,22]. It was also hypothesized that the higher rate of postoperative complications may have a negative impact on long-term outcomes as measured in lower OS and DFS rates, potentially secondary to delaying adjuvant chemotherapy [23].

One meta-analysis that included 12,817 patients with CLMs, irrespective of the timing of their development and their resection, showed a negative impact of postoperative complications on both survival and recurrence after CLM resection [24]. Of note, the meta-regression for OS revealed that the variables included in the analysis had a significant influence on the relationship between postoperative complications and OS, and none of the included studies reported all the factors that might independently influence survival [24]. Other studies revealed that the impact of postoperative complications on long-term outcomes after CLM resection depends on the type of complications. Thus, a retrospective study of 254 consecutive liver resections for CLMs proved that the detrimental oncological impact of postoperative complications is determined only by infective etiology and not the severity of complications [25]. A similar conclusion of a significant impact of infectious complications on DFS and OS resulted from a propensity score matching analysis using 2281 hepatectomies coming from a multicentric, retrospective cohort [26].

However, none of the above-mentioned studies included only the patients who underwent SR for SCLMs. The current study is the first to assess the impact of postoperative complications on long-term outcomes in a cohort of almost 250 consecutive patients with SCLMs, who underwent SR, in a single center. The results of the present study revealed that postoperative complications were not associated with significantly lower OS or DFS rates. Similarly, a small study published in 2012 evaluated determinants of short- and long-term outcomes in 46 patients who underwent SR of colorectal cancer and SCLMs and failed to demonstrate that postoperative complications have a significant impact on OS or DFS [27]. A possible explanation for the lack of impact of complications on survival rates may be the prevalence of grade I–IIIA complications (more than 80% of all complications) in the current study. Similar findings were reported in another study on 140 patients with SCLMs (treated either by SR or staged resection), which revealed that only major postoperative complications (Clavien–Dindo III–V) were an independent prognostic factor for poor OS [28]. The lack of impact of major morbidity on survival in our study could be the effect of both a relatively low rate of grade IIIB–IVB complications and the exclusion from survival analysis of the patients who died postoperatively. The reason for the exclusion of these patients is the fact that they did not die as a consequence of their malignant disease progression [29,30,31,32,33]. The relatively low incidence of grade IIIB–IVB complications in the current study is probably due to an increased rate of minor liver resections (88% of patients), which usually do not significantly raise the risk of major morbidity when they are associated with colorectal resections [21,34]. The negative impact of major liver resections on postoperative complications is suggested by the more frequent association of major hepatectomies with anastomotic fistula, compared to minor hepatectomies (29.1% vs. 13.7%). This very high incidence of anastomotic leak was observed although major hepatectomy has been performed before colorectal resection in order to avoid Pringle maneuver-associated bowel edema and ischemia (the Pringle maneuver itself being avoided in almost 60% cases). Furthermore, except for one, all rectal resections included a stoma formation (either permanent or temporary). These results led us to avoid the simultaneous performance of colorectal resection and major hepatectomy, in line with the recommendations of other centers [35,36,37,38].

Furthermore, grade I–IIIA Clavien–Dindo complications do not require reoperations and do not significantly prolong hospitalization. The rapid recovery after such complications does not postpone postoperative chemotherapy, allowing its initiation after a mean period of 6.74 weeks. This also might explain the lack of impact of postoperative complications on long-term outcomes in our study. Thus, a recent meta-analysis revealed that a significant survival benefit was achieved in patients with colorectal cancer in whom adjuvant chemotherapy started within 6–8 weeks after surgery [39].

Another potential explanation for the findings of the current study could be the fact that these patients with synchronous CLMs have a more aggressive tumor biology than those with metachronous metastases, which might mitigate the impact of postoperative complications on survival [4,5,6]. This hypothesis could be supported by the observation of the Memorial Sloan-Kettering Cancer Center group, who revealed that morbidity was not an independent predictor of either OS or DFS in patients with high clinical risk scores (CRS > 2) [40].

Although the results of this study are surprising, two recent single-center studies revealed that neither all postoperative complications nor major complications had an influence on the OS of patients treated for CLMs [29,30].

The independent prognostic factors identified in the present study were similar to those already reported in the literature [5,6,7]. Thus, N2 stage is a well-known predictor for poor OS and DFS [27,41], while the multiple and/or large metastases have already been reported as predictors of survival after SR of SCLMs [41,42,43]. Similar to the previously mentioned papers, the present study revealed that the presence of four or more SCLMs was independently associated with a poor DFS. In previous studies, the cut-off value of the maximum size of SCLMs that was correlated with survival ranged between 3 and 5 cm [30,43,44]. However, in the current study, the size of SCLMs has not been identified as an independent prognostic factor either for OS or DFS. A possible explanation for the lack of impact of tumor size on long-term outcomes in the current study might be the relatively low median size of SCLMs in this cohort (2.5 cm). Similarly, the median TBS was relatively low (3.2). Undoubtably, these findings are consequences of the policy of our center to avoid simultaneous performance of a major hepatectomy and primary tumor resection. As major hepatectomies are usually required by the presence of large liver metastases, the present series included a small number of patients with high metastatic burden. This is undoubtedly a limitation of this study, whose results cannot be extrapolated to the entire population of patients with SCLMs.

Beyond the above-mentioned limitation of the present study, there are also other limitations that one should consider. One weakness of the present study is its retrospective nature. As a consequence, some data are missing for more than 25% of this study population (CEA and CA 19-9 levels, mutational profiles, and data on postoperative chemotherapy) and therefore these analyses were not performed. However, the lack of these data has little impact on the topic of this paper, which deals with the impact of postoperative complications on OS and DFS. Another shortcoming of the study is that patient selection for SR was decided individually for each patient and not using a strict protocol and, therefore, further guidance is difficult to provide. However, being a single-center study, the decision on timing of SCLM resection was not as varied as that encountered in multicentric retrospective studies. Another potential limitation is the inclusion of patients treated over a long period of time (17 years), during which the medical oncology therapy evolved. However, this fact has a limited impact on the comparison between the long-term outcomes of patients who developed postoperative complications and those of patients who did not, as long as all the patients operated during the same period received the same oncologic therapy, irrespective their early postoperative course. Furthermore, the chemotherapy used in metastatic colorectal cancer has not changed dramatically since 2002. The most frequently used regimens include the association of 5-fluorouracil or capecitabine with either oxaliplatin or irinotecan. The major progress is the advent of anti-VEGF and anti-EGFR agents during the last decade of the study, but these drugs are used only after disease recurrence and not in the adjuvant or neo-adjuvant setting for patients with resectable SCLMs. Despite these inherent limitations, this is one of the largest single-center series evaluating the impact of postoperative complications on the long-term outcomes of patients who underwent SR for SCLMs. The findings presented here appear to justify future controlled studies (performed on larger numbers of patients with more clinical data points) that will hopefully improve the management of this important patient population.

## 5. Conclusions

Neither global morbidity nor septic or major-grade complications had an impact on the long-term outcomes of patients who underwent SR for SCLMs in a high-volume center. SR is mainly recommended in patients requiring minor hepatectomy associated with colorectal resection for uncomplicated primary tumors. These results cannot be extrapolated to patients undergoing simultaneous major hepatectomy and primary tumor resection. N2 stage of the primary tumor was the only independent prognostic factor for OS, while N2 stage and four or more SCLMs were independently associated with DFS.

## Figures and Tables

**Figure 1 healthcare-10-01573-f001:**
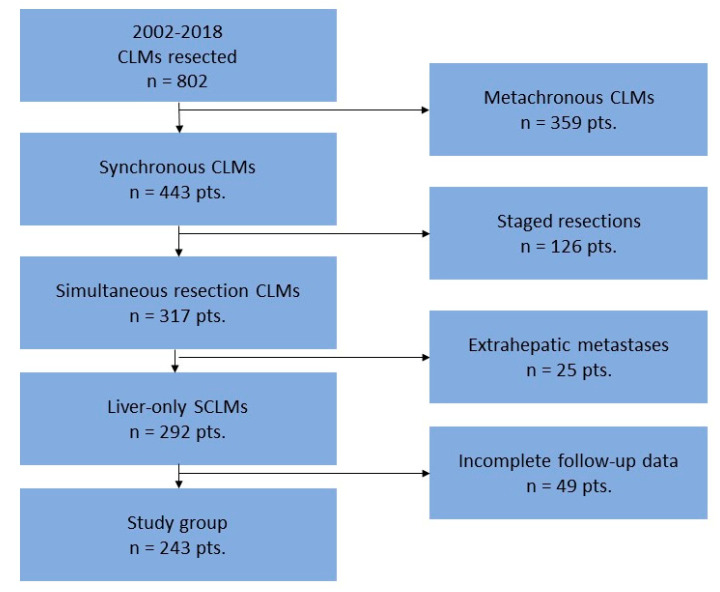
Flow-chart depicting the inclusion of the patients in the current study.

**Figure 2 healthcare-10-01573-f002:**
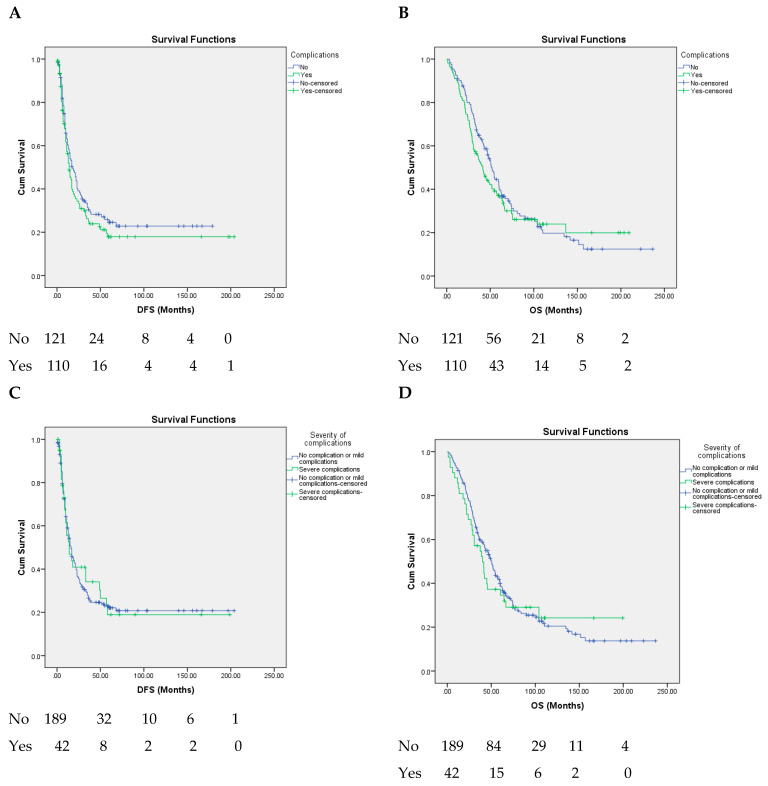
Kaplan–Meier curve of with SCLMs resected comparing (**A**) disease-free survival for complications vs. no complications, *p* = 0.245; (**B**) overall survival for complications vs. no complications, *p* = 0.488; (**C**) disease-free survival for patients without complications or with mild complications vs. severe complications, *p* = 0.907; (**D**) overall survival for patients without complications or with mild complications vs. severe complications, *p* = 0.727; (**E**) disease-free survival for septic complications vs. no septic complications, *p* = 0.661; (**F**) overall survival for septic complications vs. no septic complications, *p* = 0.391.

**Table 1 healthcare-10-01573-t001:** List of detailed procedures for the colorectal tumors.

**Procedures with Anastomosis (n = 184)**	**Number (%)**
Left colectomy	67 (27.6%)
Right colectomy	41 (16.9%)
Low anterior resection with colorectostomy and diverting ileostomy	31 (12.8%)
Dixon operation	20 (8.2%)
Subtotal colectomy	19 (7.8%)
Transverse colectomy	5 (2.1%)
Total colectomy	1 (0.4%)
**Procedures without anastomosis (n = 59)**	**Number (%)**
Colorectal resection with colostomy (Hartmann operation)	31 (12.8%)
Abdominoperineal resection	26 (10.7%)
Total pelvic exenteration	1 (0.4%)
Proctocolectomy	1 (0.4%)

**Table 2 healthcare-10-01573-t002:** Postoperative complications according to Clavien–Dindo grading.

Postoperative Complications	Mild Complications (Grade I, II) Nr. of Complications (%)	Severe Complications (Grade III, IV, V) Nr. of Complications (%)	Decease (Grade V) n (%)
**Surgical complications (n = 108)**			
Liver cut-surface collection	16 (23.53%)	24 (44.44%)	1 (8.33%)
Biliary fistula	14 (20.59%)	4 (7.41%)	0
Anastomotic fistula	13 (19.12%)	16 (29.63%)	3 (25%)
Wound complications	11 (16.18%)	3 (5.55%)	0
Pelvic abscess	5 (7.35%)	0	0
Intraabdominal abscess	0	1 (1.85%)	0
Partial bowel necrosis	0	1 (1.85%)	0
**Medical complications (n = 25)**			
Pulmonary complications	4 (5.88%)	8 (14.81%)	4 (33.33%)
Renal complications	3 (4.41%)	1 (1.85%)	1 (8.33%)
Digestive complications	2 (2.94%)	2 (3.70%)	1 (8.33%)
Cardiac complications	0	2 (3.70%)	1 (8.33%)
Clostridium difficile	2 (2.94%)	1 (1.85%)	1 (8.33%)
**Comprehensive complication index [median (IQR)]**	20.90 (8.70–20.90)	37.25 (26.20–60.57)	100
**Total number of patients**	68 *	54 **	12 ***

* Two patients had 2 concomitant complications which evolved as mild complications. ** Nine patients had 2 concomitant complications which evolved as severe complications. *** The deceased patients have also been included in the severe complications group.

**Table 3 healthcare-10-01573-t003:** Univariate and multivariate analysis of prognostic factors for DFS.

	Univariate Analysis			Multivariate Analysis		
Variable	HR	95% CI	*p* Value	HR	95% CI	*p* Value
**Age**						
≤65 y-o	1	-	-			
>65 y-o	0.932	0.673–1.289	0.668			
**Gender**						
Male	1	-	-			
Female	0.853	0.619–1.175	0.330			
**Primary tumor location**						
Colon	1	-	-			
Rectum	0.885	0.637–1.229	0.466			
**T stage**						
T1–T3	1	-	-			
T4	1.045	0.592–1.844	0.878			
**N stage**						
N0 or N1	1	-	-	1	-	-
N2	1.494	1.082–2.064	*0.015*	1.466	1.061–2.026	*0.020*
**Distributions of SCLMs**						
Unilobar	1	-	-	1	-	-
Bilobar	1.726	1.238–2.405	*0.001*	1.237	0.803–1.905	0.334
**Number of SCLMs**						
1–3 lesions	1	-	-	1	-	-
≥4 lesions	2.274	1.513–3.419	*0.001*	2.177	1.438–3.294	*<0.001*
**Size of SCLMs**						
<3 cm	1	-	-			
≥3 cm	1.243	0.911–1.696	0.171			
**TBS score**						
≤3.2	1	-	-	1	-	-
>3.2	1.411	1.032–1.929	*0.031*	1.142	0.795–1.642	0.472
**Type of hepatectomy**						
Anatomic	1	-	-			
Non-anatomic	0.905	0.594–1.378	0.640			
**Extension of hepatectomy**						
Minor	1	-	-	1	-	-
Major	1.518	0.990–2.329	0.056	0.975	0.582–1.635	0.924
**All postoperative complications**						
No	1	-	-			
Yes	1.199	0.878–1.637	0.254			
**Clavien–Dindo severe complications**						
No	1	-	-			
Yes	0.976	0.645–1.477	0.909			
**Septic complications**						
No	1	-	-			
Yes	1.075	0.773–1.496	0.667			
**Hepatic complications**						
No	1	-	-			
Yes	1.038	0.723–1.489	0.842			
**Hepatic septic complications**						
No	1	-	-			
Yes	0.988	0.649–1.504	0.954			
**Comprehensive complication index (CCI)**	1.007	0.997–1.017	0.186			
**Neoadjuvant treatment**						
No	1	-	-	1	-	-
Yes	1.562	0.994–2.456	0.053	1.488	0.945–2.345	0.086
**Adjuvant treatment**						
Yes	1	-	-			
No	0.504	0.187–1.361	0.176			

**Table 4 healthcare-10-01573-t004:** Univariate and multivariate analysis of prognostic factors for OS.

	Univariate Analysis			Multivariate Analysis		
Variable	HR	95% CI	*p* Value	HR	95% CI	*p* Value
**Age**						
≤65 y-o	1	-	-			
>65 y-o	1.255	0.919–1.714	0.153			
**Gender**						
Male	1	-	-			
Female	1.031	0.760–1.398	0.847			
**Primary tumor location**						
Colon	1	-	-			
Rectum	1.242	0.912–1.692	0.168			
**T stage**						
T1–T3	1	-	-			
T4	1.010	0.584–1.747	0.970			
**N stage**						
N0 or N1	1	-	-	1	-	** *-* **
N2	1.652	1.208–2.260	0.002	1.610	1.175–2.206	** *0.003* **
**Distributions of SCLMs**						
Unilobar	1	-	-	1	-	-
Bilobar	1.391	0.995–1.943	0.054	1.127	0.744–1.708	0.573
**Number of CLMs**						
1–3 lesions	1	-	-	1	-	-
≥4 lesions	1.573	1.023–2.418	0.039	1.490	0.960–2.314	0.076
**Size of CLMs**						
<3 cm	1	-	-			
≥3 cm	1.047	0.774–1.416	0.767			
**TBS score**						
≤3.2	1		-			
>3.2	1.209	0.894–1.636	0.219			
**Type of hepatectomy**						
Anatomic	1	-	-			
Non-anatomic	0.960	0.628–1.469	0.851			
**Extension of hepatectomy**						
Minor	1	-	-			
Major	1.080	0.676–1.725	0.747			
**All postoperative complications**						
No	1	-	-			
Yes	1.113	0.822–1.506	0.488			
**Clavien–Dindo severe complications**						
No	1	-	-			
Yes	1.073	0.723–1.592	0.727			
**Septic complications**						
No	1	-	-			
Yes	1.149	0.837–1.577	0.391			
**Hepatic complications**						
No	1	-	-			
Yes	1.099	0.768–1.573	0.604			
**Hepatic septic complications**						
No	1	-	-			
Yes	1.084	0.726–1.617	0.694			
**Comprehensive complication index (CCI)**	1.006	0.996–1.016	0.243			
**Neoadjuvant treatment**						
Yes	1	-	*-*	1	-	-
No	0.595	0.389–0.913	*0.017*	0.660	0.429–1.015	0.058
**Adjuvant treatment**						
Yes	1	-	-			
No	1.521	0.895–2.587	0.121			

## Data Availability

Not applicable.

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
