# Peer review of "The Impact of Postoperative Complications on Survival after Simultaneous Resection of Colorectal Cancer and Liver Metastases"

_healthcare, 2022, doi:10.3390/healthcare10081573_

Round 1
Reviewer 1 Report
The manuscript “The impact of postoperative complications on long-term out-comes in patients who underwent simultaneous resection of primary colorectal cancer and synchronous liver metastases: a single-center analysis of 243 consecutive patients” is a clinical manuscript, wherein 243 patients were screened. Please find my recommendation to improve the manuscript.
1. The title is very long and must be shortened.
2. In the introduction, the authors reported that “Approximately one-quarter of colorectal patients are diagnosed with synchronous liver metastasis (SCLMs), while 30% develop liver metastases subsequently (metachronous metastases)”. It will be interesting if the gene responsible for this is discussed in the introduction. A study https://doi.org/10.1002/mgg3.910, reported that Bcl-2 mutations are associated with 33 cancer types.
3. Authors should discuss, why Fischer’s exact is important for this study? Why not a chi-squared test?
4. HR is reported, but the ODD ratio (OR) should also be calculated for complete analysis.
5. Authors should also provide cohort statistical analysis of screened patients in the form of a flow diagram.
Author Response
Dear Reviewer,
Thank you very much for your comments and recommendations aiming to improve the quality of our manuscript! Please find bellow the answers to your questions, comments and recommendations.
- The title is very long and must be shortened.
We changed the title. The actual title is “The impact of postoperative complications on survival after simultaneous resection of colorectal cancer and liver metastases”.
- In the introduction, the authors reported that “Approximately one-quarter of colorectal patients are diagnosed with synchronous liver metastasis (SCLMs), while 30% develop liver metastases subsequently (metachronous metastases)”. It will be interesting if the gene responsible for this is discussed in the introduction. A study https://doi.org/10.1002/mgg3.910, reported that Bcl-2 mutations are associated with 33 cancer types.
In the Introduction we added the paragraph: “Among the genetic mechanisms involved in the progression of colorectal cancer, Bcl-2 – Bax binding sems to play an important role, through the inhibition of apoptosis. Thus, Bax mutations would be a target to prevent the progression of primary colorectal cancers to metastatic stage [3].”.
We also added a new reference: 3. Pawan Kumar Raghav,Rajesh Kumar,Vinod Kumar,Gajendra P. S. Raghava. Docking-based approach for identification of mutations that disrupt binding between Bcl-2 and Bax proteins: Inducing apoptosis in cancer cells. Mol Genet Genomic Med. 2019;7:e910. PMID: 31490001 DOI: 10.1002/mgg3.910
- Authors should discuss, why Fischer’s exact is important for this study? Why not a chi-squared test?
In the Materials and methods section (2.6. Statistical analysis) we added a new paragraph: “Fisher’s exact test is preferable because it is an exact test (while the chi-squared test relies on an approximation) and it is more suitable in analysis of small samples [15].”
We also added a new reference: 15. Hae-Young Kim. Statistical notes for clinical researchers: Chi-squared test and Fisher’s exact test. Restor Dent Endod 2017;42:152-5. PMID: 28503482 DOI: 10.5395/rde.2017.42.2.152
- HR is reported, but the ODD ratio (OR) should also be calculated for complete analysis.
We used HR because it allows a measure of association in outcomes in survivorship curves. According to Andrew George, Thor S. Stead and Latha Gant “both RR and OR concern interventions and outcomes, thus reporting across an entire study period. However, a similar but distinct measure, the hazard ratio (HR), concerns rates of change. HRs are in tandem with survivorship curves, which show the temporal progression of some event within a group, whether that event is death, or contracting a disease. In a survivorship curve, the vertical axis corresponds to the event of interest and the horizontal axis corresponds to time. The hazard of the event is then equivalent to the slope of the graph, or the events per time.” The authors stated that “OR is static – does not consider rates; summarizes an overall study. HR is based on rates; provides information about the way a study progresses over time”.
George A, Stead T S, Ganti L (August 26, 2020) What’s the Risk: Differentiating Risk Ratios, Odds Ratios, and Hazard Ratios?. Cureus 12(8): e10047. DOI 10.7759/cureus.10047 https://www.ncbi.nlm.nih.gov/pmc/articles/PMC7515812/pdf/cureus-0012-00000010047.pdf
- Authors should also provide cohort statistical analysis of screened patients in the form of a flow diagram.
We attached the flow-chart diagram (Figure 1)
Figure 1. Flow-chart depicting the inclusion of the patients in the current study.
Thank you very much for your recommendations!
Sincerely yours,
Narcis Octavian Zarnescu
Reviewer 2 Report
Thank you for your interesting study
But you have to consider the following points:
Abstract
1. You have a missing verb in the first statement of the abstract “ The aim of this to investigate “ it should be “ the aim of this study was to investigate ….”
Introduction
1. You need to illustrate the significance of the study “ you can add more studies to clarify the uniqueness or the contribution of such study”. The gap in literature should be clarified.
Study design and population
1. The design of the study is not mentioned and described.
2. The author should provide more information about the selected sample and the recruitment process). In addition, further information is needed to ensure that survival rates are not affected by other extraneous factors such as other morbid diseases.
3. The eligibility criteria were illustrated by the authors but (it is better to base this inclusion and exclusion criteria on a solid literature).
Outcome measurement
1. The researchers mentioned that the study approved by local IRB but they didn’t illustrate and explain how they ensure anonymity and confidentiality of the patient during data collection and follow up period.
2. It is better to provide a diagram that illustrate the data collection process (indicating different stages at which data are collected and study variables)
Author Response
Dear Reviewer,
Thank you very much for your comments and recommendations aiming to improve the quality of our manuscript! Please find bellow the answers to your questions, comments, and recommendations.
Abstract
- You have a missing verb in the first statement of the abstract “The aim of this to investigate“ it should be “the aim of this study was to investigate ….”
The modification was made. “The aim of this study was to investigate the impact of postoperative complications on long-term outcomes of patients who had undergone simultaneous resection (SR) of colorectal cancer”
Introduction
- You need to illustrate the significance of the study “you can add more studies to clarify the uniqueness or the contribution of such study”. The gap in literature should be clarified.
We have added a new paragraph in the Introduction section:
“A major concern regarding SR is the higher incidence of postoperative complications and their potential impact on long term outcomes and this has led to reluctance for performing simultaneous resection [13,14]. Despite the attractiveness of the concept of simultaneous excision of colon cancer and liver metastases, there is still a paucity of high-quality evidence (randomized clinical trial). This study will evaluate the outcomes of simultaneous colorectal and liver resection utilizing a pragmatic approach and rigorous patient selection in a high-volume center for both procedures.”
Furthermore, we added 2 new references:
- Serrano PE, Parpia S, Karanicolas P, Gallinger S, Wei AC, Simunovic M, et al. Simultaneous resection for synchronous colorectal cancer liver metastases: A feasibility clinical trial. J Surg Oncol 2022;125:671-7. PMID: 34878649 DOI: 10.1002/jso.26764.
- Chen Q, Deng Y, Chen J, Zhao J, Bi X, Zhou J, et al. Impact of Postoperative Infectious Complications on Long-Term Outcomes for Patients Undergoing Simultaneous Resection for Colorectal Cancer Liver Metastases: A Propensity Score Matching Analysis. Front Oncol 2021;11:793653. PMID: 35071001 DOI: 10.3389/fonc.2021.793653.
Study design and population
- The design of the study is not mentioned and described.
This is a survival cohort study.
We have introduced a description of the study design in the Material and methods section, subchapter “2.1. Study design and population” (see below).
- The author should provide more information about the selected sample and the recruitment process). In addition, further information is needed to ensure that survival rates are not affected by other extraneous factors such as other morbid diseases.
The issue of comorbidities is presented in the subchapter “2.2. Treatment allocation” (see the answer to the next comment).
- The eligibility criteria were illustrated by the authors but (it is better to base this inclusion and exclusion criteria on a solid literature).
Regarding the recruitment process and eligibility, as mentioned in manuscript, we have used a prospectively maintained database including all the patients who underwent resection of colorectal liver metastases in our center since 2012. The patients with metachronous metastases were removed, thus collecting for this study all consecutive patients with synchronous colorectal cancer and liver metastases. We have also excluded patients that undergone staged resections of primary colorectal tumor and liver metastases, patients with extra-hepatic metastases and those with incomplete follow-up data. However, there was no sub-selection patient-process based on medical comorbidities. We re-done the subchapter “2.1. Study design and population” and added a flow-chart diagram (Figure 1):
“2.1. Study design and population
We conducted a single-institution survival cohort study in patients with SR, collecting clinical, pathological, and postoperative complications data. All consecutive adult patients who underwent SR between January 2002 and December 2018 were retrieved from a prospectively maintained database including all the patients who underwent resection of CLMs in our center. Patients with extra-hepatic metastases, as well as those with incomplete follow-up data were excluded from this analysis. The flow-chart diagram of the included patients is presented in Figure 1.”
The eligibility criteria are presented in the subchapter “2.2. Treatment allocation”, along with references supporting these criteria.
“2.2. Treatment allocation
Patients with SCLMs were usually evaluated by a multidisciplinary team (including oncologists, radiologists, as well as colorectal and hepatic surgeons). Multidisciplinary team has recommended the timing of multimodal treatment and surgical team selected patients for surgical procedure.
The decision to perform SR or staged resection was made by the attending surgeon based on preoperative and intraoperative findings (complexity of the hepatectomy and colectomy), performance status of the patient and the anticipated technical ability to per-form a complete resection of the primary tumor and SCLMs with preservation of more than 30% of total liver volume. In line with the current practice, the policy of the center is to perform SR in patients with uncomplicated primary tumors (no bowel occlusion or perforation) and requiring minor hepatectomies [15,16]. Major liver resections have been considered only in patients with very good performance status and who did not undergo a difficult colorectal procedure.
As a principle, patients with severe medical comorbidities were excluded from simultaneous resection. In recent practice survey, patient comorbidities were ranked as the most significant barrier by 35% of surgeons [16]. Furthermore, performance status is used to guide patient selection for simultaneous resection [15]”
We also added 2 new references:
- Kleive D, Aas E, Angelsen JH, Bringeland EA, Nesbakken A, Nymo LS, Simultaneous Resection of Primary Colorectal Cancer and Synchronous Liver Metastases: Contemporary Practice, Evidence and Knowledge Gaps. Oncol Ther 2021;9:111-20. PMID: 33759076 DOI: 10.1007/s40487-021-00148-2.
- Griffiths C, Bogach J, Simunovic M, Parpia S, Ruo L, Hallet J, Serrano PE. Simultaneous resection of colorectal cancer with synchronous liver metastases; a practice survey. HPB (Oxford) 2020;22:728-34. PMID: 31601509 DOI: 10.1016/j.hpb.2019.09.012
Outcome measurement
- The researchers mentioned that the study approved by local IRB but they didn’t illustrate and explain how they ensure anonymity and confidentiality of the patient during data collection and follow up period.
We keep our records safe by using password-protected files. Furthermore, we do not record information in such a way that subject responses are linked to identifying information (names, addresses, e-mail addresses, phone numbers).
- It is better to provide a diagram that illustrate the data collection process (indicating different stages at which data are collected and study variables)
We attached the flow-chart diagram (Figure 1):
“Figure 1. Flow-chart depicting the inclusion of the patients in the current study”
Thank you for your recommendations!
Sincerely yours,
Narcis Octavian Zarnescu